# Implementation of good clinical practice in clinical research in the context of limited resources settings: Lessons learnt from the freeBILy trial using an embedded mixed methods approach

Leonard Gunga[1,2☯], Pia Rausche[1,2☯], Rivo Andry Rakotoarivelo[3], Mandranto Rasamoelina[4], Jeannine Solonirina[5], Elveric Fesia[3], Ravo Razafindrakoto[4], Njary Rakotozandrindrainy[5], Mickael Radomanana[3], Valentina Marchese[1,2], Nagham Issa[1], Jean-Marc Kutz[1], Aaron Remkes[1,2], Anna Jager[2,6], Dewi Ismajani Puraduredja[7], Govert van Dam[8], Norbert Schwarz[6], Jürgen May[2,6], Raphaël Rakotozandrindrainy[5], Natalie Fischer[1‡], Daniela Fusco[1,2‡*] on behalf of the freeBILy consortium[¶]

1 RG Implementation Research, Bernhard Nocht Institute for Tropical Medicine (BNITM), Hamburg, Germany, 2 German Center for Infection Research (DZIF), Hamburg-Borstel-Lübeck-Riems, Hamburg, Germany, 3 Infectious diseases service, University Hospital Tambohobe, Fianarantsoa, Madagascar, 4 Centre Infectiolologie Charles Mérieux (CICM), Antananarivo, Madagascar, 5 University Antananarivo, Antananarivo, Madagascar, 6 Department of Infectious Diseases Epidemiology, Bernhard Nocht Institute for Tropical Medicine (BNITM), Hamburg, Germany, 7 Interdisciplinary Academy of Competence and Education for Global Health (iACE), Bernhard Nocht Institute for Tropical Medicine (BNITM), Hamburg, Germany, 8 Department of Parasitology, Leiden University Medical Center, Leiden, the Netherlands

¶ Membership of freeBILy consortium is provided in the Acknowledgements.
☯ These authors contributed equally to this work.
‡ NF and DF also contributed equally to this work.
* fusco@bnitm.de

## Abstract

### Introduction

Limited financial and human resources and infrastructure can affect the implementation of Good Clinical Practice (GCP), which can have a detrimental impact on data quality and the robustness and application of clinical trial outcomes. Monitoring frameworks are designed to ensure good data quality and help to guide adaptations of trial procedures over time. However, these frameworks tend to be based on datacentric approaches, which often neglect vital aspects of trials, such as social responsibility, capacity strengthening, and contextual influence. Therefore, this study analyses barriers and facilitators of the implementation of GCP in resource-limited settings to inform the establishment of adapted frameworks for trial management and monitoring.

**Data availability statement:** The datasets used and analyzed during the current study have been uploaded as Supporting information together with the data dictionary, the transcriptions of qualitative are available from the BNITM data service (data.requests@bnitm.de) on reasonable request and after validation of the freeBILy steering committee.

**Funding:** This publication was produced by freeBILy which is part of the EDCTP2 program supported by the European Union (grant number RIA2016MC-1626-FREEBILY to GvD). The views and opinions of authors expressed herein do not necessarily state or reflect those of EDCTP. The funding body had no role in the design of the study; collection, analysis and interpretation of the data; and writing of the manuscript. JMK received support through the Bayer Foundation (Carl Duisberg Fellowship for Medical Sciences). Open Access funding enabled and organized by Projekt DEAL.

**Competing interests:** The authors have declared that no competing interests exist.

## Methods

In this multi-method analysis of the freeBILy trial, conducted in Madagascar from 2019-2022, a random subset of trial participants (n=500) and informed consents (n=500) was analyzed for protocol deviations through descriptive statistics and trend analysis. Framework analysis of focus group discussions and individual semi-structures interviews provided a sociological viewpoint of the study context. Findings were subsequently triangulated, merging the viewpoints on the influences towards GCP in resource-limited settings.

## Results

A decreasing trend in incorrect database entries was found (z=-6.968, Mann-Kendall Test, p<0.001) over the course of the study, with an overall rate of 1.8% incorrect data entries. Triangulation showed three key areas of GCP implementation in resource-limited settings, which extend previous frameworks: a) Context adaptation towards infra-, team- and social structures as promoting factors, b) External influences, such as external threats, study personnel attitudes and perception towards the trial require recurrent assessment, and c) Promote GCP-compliant data collection subject to regular documentation and training cycles to facilitate capacity strenghtning and data ownership.

## Conclusion

This study shows the limitations of datacentric clinical trial management to assess GCP performances in the frame of clinical trials in resource-limited settings. We highlight the importance of well-trained and integrated study staff, as well as thorough preparation, budgeting and context appropriate monitoring. This achieves high quality, patient centered and compliant research, implemented through alternative frameworks for monitoring and evaluation.

### Author summary

This work has the scope to advise and advance practices for implementation of clinical trials in resources limited settings and particularly in sub-Saharan Africa. The study was implemented in the context of a big clinical trial, freeBILy, conducted in three rural regions of Madagascar and fully embedded into the ante natal care services of the country. Our findings encourage to adapt monitoring practices for clinical trials in order to guarantee reliable data to support the knowledge advancement for infectious diseases of neglected populations.

### Introduction

Clinical trials to address diseases of poverty, such as malaria and neglected tropical diseases (NTDs) have increased in the last decade [1]. Nevertheless, the implementation of such studies in endemic contexts can pose challenges. Limited financial

and human resources and infrastructure can have a detrimental impact on the implementation of Good Clinical Practice (GCP), and potentially lead to poor data quality that will fail to provide adequate clinical and policy guidance [2]. Here, we refer to "GCP" in alignment with the Guidelines of the Harmonization of Technical Requirements for Pharmaceuticals for Human Use (ICH), at the time of the implementation of this trial E6 (R2) from 2016 [3].

The ICH GCP originated from the context of the EU, US and Japan. Several researchers suggested that careful contextualization is needed to enable adherence to GCP in resource-limited settings [4]. In particular, it has been argued that having been developed for high-income contexts and with a strictly regulatory approach, the ICH GCP failed to address essential elements for collaborative research in global health. For example, the implementation of social responsibility, capacity strengthening, benefit sharing and contextual impact on the study participants and communities [5]. In the past, many GCP training programs and monitoring did not account for these aspects, despite the evident need for context sensitive approaches [6]. The importance of a shift to a process-based clinical trial management is also reflected in the recently revised E6 (R3) guideline [7]. Furthermore, the involvement of local stakeholders in the planning of clinical trials has been emphazised building on the concepts of E8 (R1) [8]. Although this represents a step in the right direction, this does not yet seem sufficient to encourage a thoughtful contextualization of trials and their management in low-resource settings as guidelines remain vague in their formulation in this regard. Additional contextualization of trials in their intended implementation setting, as an element of careful trial impact conceptualisation and the fit for purpose concept, is urgently needed [9]. It is necessary to take the characteristics and constraints of the local health systems into account, and to ensure the conduct of clinical trials vis-a-vis unexpected external threats such as outbreaks, political instability, natural disasters or public health emergencies among others [10].

Many researchers use checklists and standardized monitoring tools developed for high income settings for the analysis of quality indicators in clinical trials in resource-limited settings [11]. However, such monitoring tools may neglect important contextual aspects that influence the scientific and ethical conduct of a trial while imposing a high burden of paperwork [12]. The main viewpoint on purely technical and number focused aspects prevents researchers to adopt a contextualized 'holistic approach', which is needed to strengthen both local infrastructure and competency to conduct clinical research and translate its findings into policies and practices.

To the best of our knowledge, there is limited evidence-based guidance using context-sensitive frameworks for improved compliance and adherence to adequate scientific and ethical standards [4,5,13].

Therefore, this study aimed to analyze barriers and facilitators for the implementation of ICH GCP in resource-limited settings, to inform the establishment of comprehensive research quality management in such contexts, evaluating lessons learnt from the freeBILy (**f**ast and **r**eliable **e**asy-to-use diagnostics for **e**liminating **bil**harzia in **y**oung children and mothers) trial conducted in Madagascar [14].

Specifically, the objectives of this study were to (i) quantify incorrect data entry fields and erroneous informed consents, (ii) identify factors influencing non-conformities (i.e., protocol deviations) in adherence to the study protocol, (iii) explore GCP-related perceptions, practices and recommendations among study personnel in Madagascar.

## Methods

### Ethical considerations

IC was obtained from all participants of the freeBILy trial and all study staff involved. All forms containing participants' signatures were kept separate from the questionnaires to protect the participants privacy and confidentiality. The study freeBILy in Madagascar was conducted in line with the ICH-GCP E6 (R2) guidelines and the principles of the declaration of Helsinki. Ethical approval was obtained by the National Ethics Committee of Madagascar (ref. no 022-SANP/CERBM of 05/03/2018) and the Ethics Committee of the Hamburg State Medical Chamber in Germany (ref. no PV5966 of 18/03/2019). The study was retrospectively registered on 15 May 2019 in the Pan-African Clinical Trial Register

(PACTR201905784271304). Written informed consent was obtained from all participants included in the study. Participants had the right to withdraw informed consent and study participation at any time without giving reasons.

## Study context

This study is a secondary analysis of data from the freeBILy trial: A two-armed, randomized phase III trial, aimed to determine the effectiveness of a test-based schistosomiasis treatment strategy for pregnant women aged 16 years and above as well as their infants followed up until 24 months after birth [14]. The trial was conducted from 2019 to 2022, in 40 primary health care centers (PHCCs) in the regions of *Bongolava*, *Itasy* and *Amoron'I Mania* in Madagascar, with high *Schistosoma mansoni* endemicity. Study staff was initially trained and retrained on the basis of the ICH-E6 (R2) guideline [3]. During the trial, measures of community engagement such as awareness initiatives and community involving side events such as the distribution of coffe plants were held to facilitate attrition of the trial. freeBILy represents a good paradigm as it was implemented in remote areas of a country were a limited number of clinical trials were implemented overall giving the possibility to asssess the validity of monitoring tools in a relatively new setting.

## Conceptual framework

The framework in which this study was conceptualized is based on the clinical trial quality model described by de Pretto-Lazarova et al. which contextualizes quality in trials in resource-limited settings in 2 main categories, namely promoting factors and building factors, which are further divided into moral and scientific factors [5] (Fig 1). This concept was developed in an investigator, monitor and sponsor focused qualitative approach. Our study provides additional insight from the study staff and actual trial implementation in relation to this conceptual framework of trial quality, and therefore ultimately GCP implementation using an embedded mixed-method design after Creswell and Plano Clark (2011). The qualitative staff interviews are added and nested within the analysis of the freeBILy database to add depth towards the understanding of barriers and facilitators of GCP in low-resource settings [15].

## Quantitative data collection and management

Three types of quantitative data were collected: entries from case report forms (CRFs), track record of informed consent (IC) and interviewer administered questionnaires among study staff.

At recruitment into the freeBILy study, each participant received a unique 12-digit Participant Identification number (PID), linking mother and child study data to their identifiers. The study visits were conducted in Malagasy, while the CRFs were in French. All participant data were recorded first in paper-based CRFs and then entered into a REDCap database, at the five study time points (T0-T4) from May 2019 to September 2022: fifth or sixth month of pregnancy (T0), eighth month of pregnancy (T1), delivery or infant enrollment (T2), nine to 12 months postpartum (T3), and 24–26 months postpartum (T4) [14,16,17].

CRFs data included demographic, anthropometric and socio-demographic information, diagnostic and laboratory test results regarding schistosomiasis and pregnancy related outcomes. Data quality checks and double data entry of paper-based CRFs into a REDCap database hosted by the Bernhard Nocht Institute for Tropical Medicine (BNITM) in Hamburg, Germany, were performed by clinical research associates in Madagascar. For the present study, track and change record data of the CRFs were extracted into a CSV spreadsheet from the database. These records included CRF fields, and records of opening and closing a specific study visit of a participant as well as corrections in the database with indication of time and responsible person.

Prior to study start, local staff was trained on study protocol and GCP in November 2018. These trainings were conducted in Malagasy and covered topics such as participant safety, adherence to the study protocol and handling of serious adverse events.

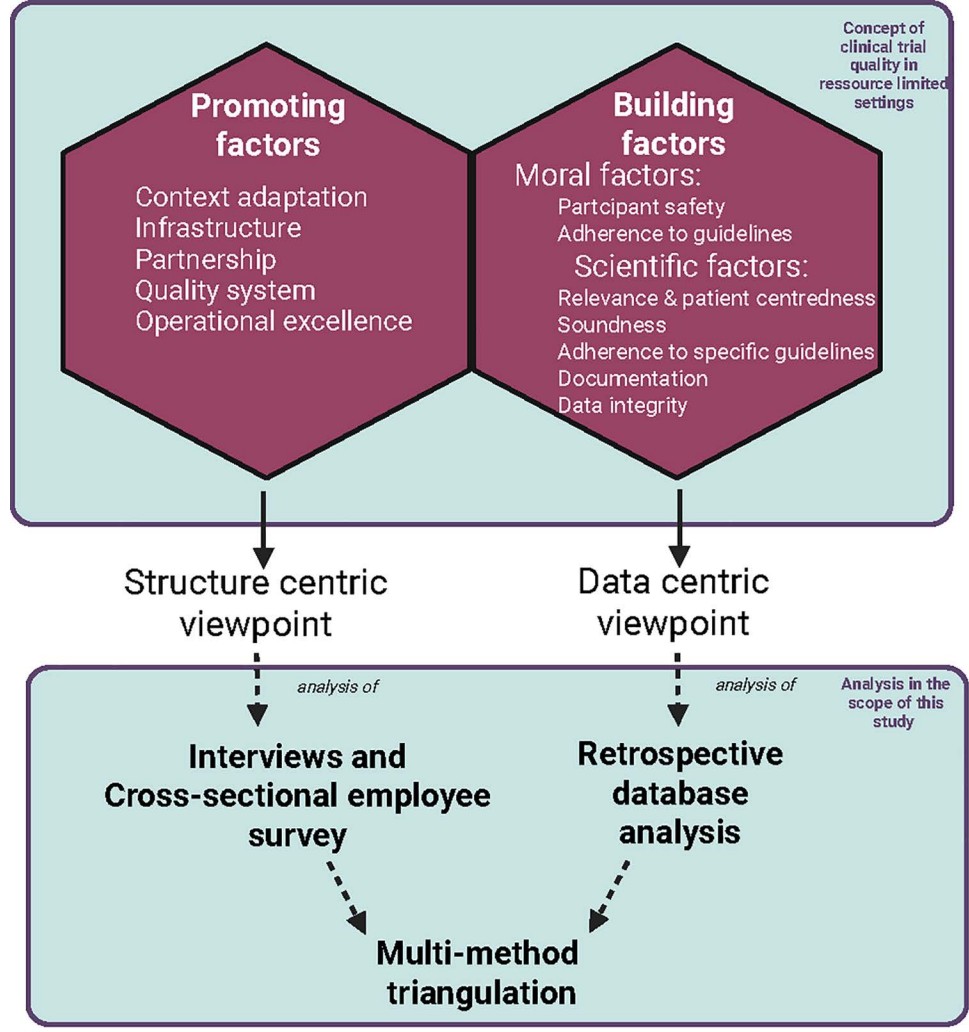

**Fig 1. Overview of conceptual framework of this study, based on the concept of clinical trial quality in resource-limited settings (box above) by de Pretto-Lazarova et al. (2022) regarding the methodology applied in the scope of this study (box below).** Created in BioRender. Fusco, D. (2025) https://BioRender.com/hquc7ry.

The IC was written in Malagasy and consisted of an information sheet and a consent sheet. In the consent sheet different tick boxes were included to allow for multiple levels of consent, and when needed, for the possibility of opt-out, for different components of the study. Additionally, a separate section allowed for guardian consent for minor participants and impartial witnesses for illiterates. During the study period, the IC was updated to a newer version, due to an amendment of the protocol (e.g., introduction of additional questionnaires), requiring re-approval of the ethics committee, and re-consent of all participants. For the purpose of this study, the ICs were manually reviewed by the study's clinical research assistant and saved in Excel format.

For the present study, a structured questionnaire was administered to study staff (e.g., laboratory personal, primary health care workers (HCWs), data entry clerks) to collect information on socio-demographic background, self-assessed language skills and GCP trainings.

## Qualitative data collection and management

Qualitative data collection was performed from August to September 2022. Individual semi-structured interviews (ISI) were conducted with 15 purposively selected employees for information rich cases. One focus group discussion (FGD) was conducted with the remaining 15 employees not yet included. ISIs and FGDs were conducted following a semi-structured, pre-tested topic guide. Key emerging elements, such as the process of obtaining IC and trial workflow were thematized. Barriers and facilitators of GCP were discussed. The participants were asked to provide recommendations for the implementation of GCP in the setting of Madagascar, in relation to the definition of GCP presented in the received ICH-GCP training.

ISIs and FGDs were conducted in Malagasy (MJS with assistance of JMK). Interviews were audio recorded, with participants' permission, with an average interview duration of 32 minutes (min. 14 minutes, max. 90 min.). They were first transcribed in verbatim (MJS), and subsequently translated from Malagasy into French and English, to allow data analysis and review from non-French speaker staff. A translation check by a native speaker (MJS: Malagasy to French and JMK: French to English) was provided. The pseudonymized transcripts of FGD and ISIs were uploaded onto the password protected secure cloud server hosted by the BNITM.

## Quantitative data analysis

Among the 5114 women recruited by freeBILy, a subset of 500 were randomly selected for this secondary analysis [18,19]. Sampling and analysis was conducted using R 4.3.1. A seed was set and 11% of participants were randomly sampled within PHCC strata, ensuring proportional representation of each PHCC. Due to group-level rounding, this yielded 500 participants (about10% of the total cohort). Analysis of data entry corrections were conducted on the merged dataset after double data entry. Multiple corrections for the same data entry field were detected and counted during the data quality assessment. Missing data entries were considered data entry errors only if they were subsequently corrected (i.e., modified from missing to complete). The proportion of data entry errors was calculated as incorrect data entries over total number of records. CRF elements were categorized into six groups: dates, binary categorical variables, multi-level categorical variables, free text, numeric variables, and study IDs [20]. Throughout the course of the study, the number of possible data entry fields in the CRF varied per timepoint, as different variables were collected throughout the study period. Incorrect data entries were normalized by the median number of entry fields per study timepoint. Data entry errors were categorized by severity [21]: critical (inclusion and exclusion criteria, participant safety or primary outcomes), moderate (health-related data not impacting safety or protocol) and minor (demographics, remarks, variables related to sub-studies). The trend of incorrect data entries per week was assessed using a Mann-Kendall test from R trend package [22].

For each of the 500 randomly selected participants, the IC was evaluated and the proportions of errors per IC field was assessed. The errors were categorized into being related to the participant, study staff, and the enrollment of minors or illiterate participants. This covered verification of names, signatures, and dates for both participants and staff, confirming the information provided by legal guardians (parents) of minors, and documenting the thumbprints for illiterate participants and signatures of impartial witnesses. It also included checking the presence of the most recent IC version with all pages and whether the IC had to be re-obtained.

## Qualitative data analysis

Qualitative data was analyzed by PR and NI using the framework analysis approach [23,24]. Data familiarization reports of ISIs and FGDs were created, followed by the identification of a thematic framework by PR and NI and indexing of the data [24]. Coding and charting were performed by PR using MAXQDA (VERBI GmbH, Berlin, Germany). A-priori themes and emerging themes were summarized and mapped in a matrix format. Example quotes for

each topic were presented in the framework matrix (S5 Table). A-priori themes included in the topic guide and the analysis included understanding and attitudes towards GCP, IC and recommendations for GCP training. Emerging themes included workflow, barriers to GCP implementation, facilitators of GCP compliance and staff motivation. Some of the themes were mentioned by a small number of participants, yet these themes were identified as crucial for the analysis or discussion shaping. Displayed quotes were labelled with a pseudonymized identifier and profession of the employee.

## Triangulation

To identify the determinants of clinical trial quality, the factors influencing it, as well as quality assurance measures in this resource-limited setting, all results from quantitative and qualitative data collections were first analyzed separately. Using an embedded, among method triangulation, where as the interviews, focus group discussions and staff survey were embedded into the context of the freeBILy study, findings were compared for the purpose of validation and interpretation using the conceptual framework by de Pretto-Lazarova et al. as described above. Results were charted in a side by side matrix display (S6 Table) and discussed among the researchers [25]. Afterwards, findings from the side by side display were translated into a visual approach.

## Results

### Evaluation of CRFs incorrect data entry fields

Among 294477 data entry fields, a total of 1.8% (n = 5211) were incorrect (Table 1). To visualize incorrect data entries over time, normalized incorrect data entries per week and external factors such as GCP trainings, monitoring visits and the COVID-19 lockdowns were mapped across the study period (Fig 2). Over the first year of the study (May 2019-May 2020), visits for T0, T1 and T2 overlapped in the PHCCs. The second year focused on T3 visits only (May 2020-May 2021), while the final year focused on T4 visits only (August 2021-September 2022). While there was no visible reduction in study visits during the first COVID-19 lockdown (March-October 2020), there was a shutdown of activities during the second lockdown. By the time the T4 visits started, two external monitoring visits and four GCP trainings had taken place. Over the entire study duration of 152 weeks, a decreasing trend in incorrect data entry fields was observed (z = -6.968, Mann-Kendall Test, p < 0.001).

The median overall number of required data entry fields per questionnaire increased from early (T0, T1, T2) to later timepoints (T3, T4) (S1 Table).

Among the incorrect entries normalized by data entry fields per study time point an increase from T0 (3.1%) to T1 (3.6%) and T2 (3.9%) was observed, followed by a decrease to T3 (2.7%) and T4 (2.8%) (S1 Table).

**Table 1. Summary of incorrect data entries categorised by critical issues related to the primary endpoint of the study.**

| Severity | Incorrect data entries, n (%) | All data entries fields n (%) | Proportion incorrect entries # (%) |
|---|---|---|---|
| Overall | 5211 (100) | 294 477 (100) | 1.8 |
| Critical | 1831 (35.1) | 120 180 (40.8) | 1.5 |
| Moderate | 2216 (42.5) | 149 838 (50.9) | 1.5 |
| Minor | 1164 (22.4) | 24 459 (8.3) | 4.8 |

# Percentages are rounded up to one decimal.

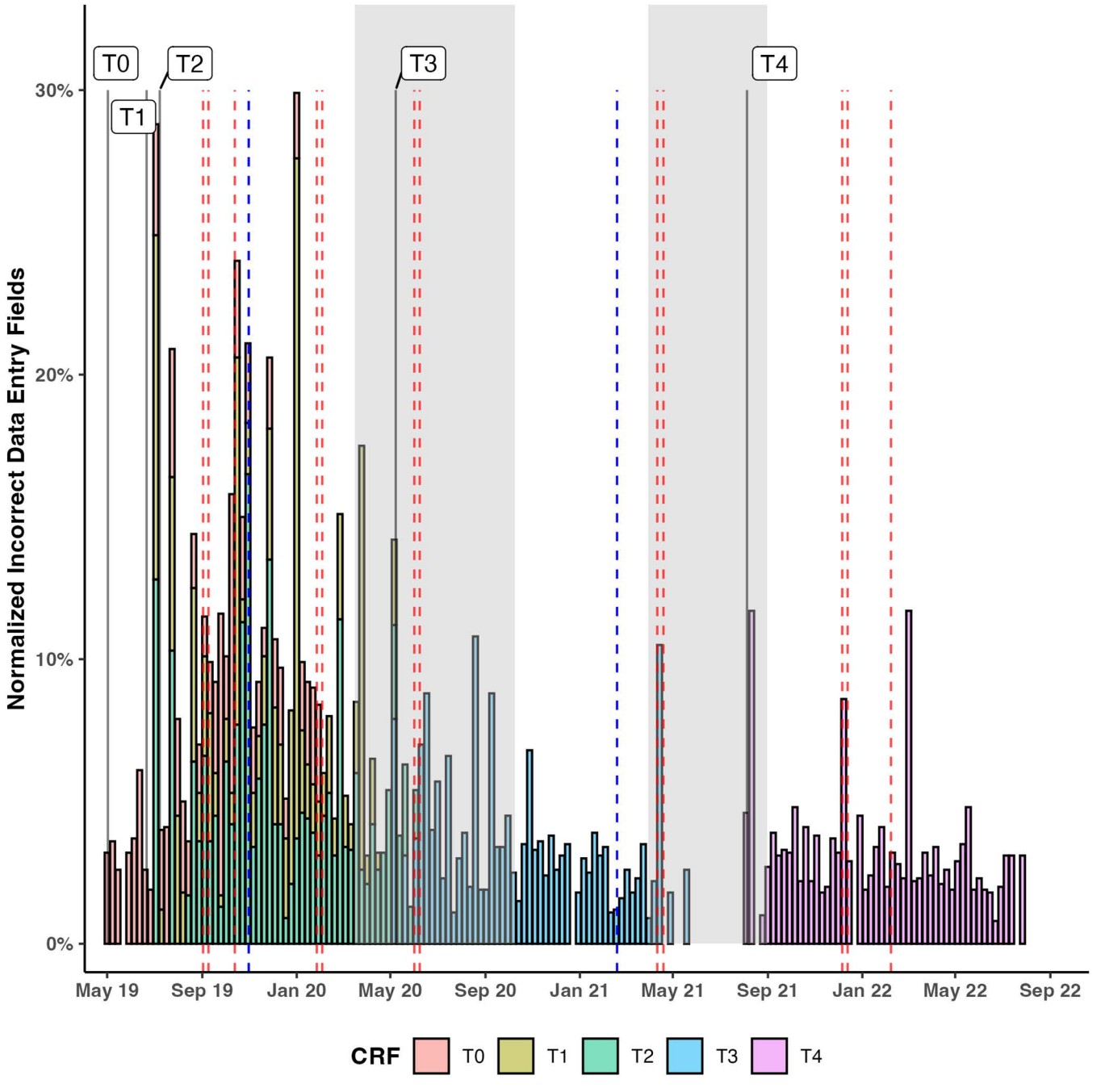

**Fig 2. Normalized CRFs incorrect data entry fields per week throughout the study: The graph shows the weekly incorrect data entry fields from May 2019 to September 2022.** The stacked colored bars (orange=T0, yellow=T1, green=T2, blue=T3, pink=T4) show the percentage of incorrect data entry fields normalized by the number of possible data entry fields per week. Vertical black lines represent the start of each of the five study time points (T0-T4). Red dashed lines illustrate the internal GCP trainings, whereas the dark blue lines represent the external monitoring visits (November 2019, on-site and February 2021, remotely due to COVID-19). The grey-shaded areas correspond to the timeframe of the COVID-19 national health emergency lockdowns.

The overall proportion of incorrect data entries for variables categorized as critical and moderate was the same (1.5%) (Table 1). The highest percentage of incorrect entries was detected for variables categorized as of minor impact (4.8%), which constituted the least of data entry fields (24259, 8.3%).

Further, the 5211 incorrect data entry fields were categorized into six distinct groups according to data type (S2 Table). The most frequent incorrect data entries were associated with dates (24.9%) (e.g., date of study timepoint visit or vaccinations of the child) and binary categorical data (24.0%) including critical variables such as the study drug administration. Both categorical variables with levels and text sections accounted for about 20% of incorrect data entries. The incorrect data entries of numeric variables, such as anthropometric measurements or diagnostic results accounted for a total of 10.4%. An incorrect entry of the participants study ID was rare (0.36%). Regarding the primary outcome variables of maternal hemoglobin at T0, T1, T3 and T4, as well as infant height and weight at T3 and T4, 1.1% (123/ 10798) of data entry fields were incorrect (S3 Table).

## Evaluation of IC errors

At the end of the study, all 500 IC forms reviewed were of the most recent EC-approved version, although in 12.8% (64/ 500) of cases an error (i.e., use of the older form) had already been corrected in the past (Table 2). The name of the participant/guardian/witness was missing or incorrect in 2.4% (24/ 1000) of cases. The date of the participant/guardian/witness signature was missing in 4.8% (24/ 500) and the participant/guardian/witness signature in 2.8% (28/ 1000) of cases.

**Table 2. Evaluation of IC errors.**

| Informed consent variables | Total possible | Yes | No | Errors (%) |
|---|---|---|---|---|
| **Participant-related variables for all reviewed ICs (N = 500)** | | | | |
| non printed name filled in* | 1000 | 976 | 24 | 2.4 |
| Participant/guardian/witness printed name correct* | 1000 | 976 | 24 | 2.4 |
| Participant/guardian/witness signature present* | 1000 | 972 | 28 | 2.8 |
| Participant/guardian/witness signature date present | 500 | 476 | 24 | 4.8 |
| Checkbox on being well informed and having received information sheet completed | 500 | 479 | 21 | 4.2 |
| **Participant-related variables for minors (N = 26/ 500, 5.2%)** | | | | |
| If minor, is printed name of guardian present | 26 | 26 | 0 | 0 |
| If minor, is signature of guardian present | 26 | 23 | 3 | 11.5 |
| **Participant-related variables for illiterates (N = 20/ 500, 4%)** | | | | |
| If illiterate, is thumbprint present* | 40 | 35 | 5 | 12.5 |
| If illiterate, is printed name of impartial witness present | 20 | 19 | 1 | 5.0 |
| If illiterate, is signature of impartial witness present | 20 | 19 | 1 | 5.0 |
| If illiterate, is date of signing present | 20 | 15 | 5 | 25 |
| **Staff related variables for all reviewed ICs (N = 500)** | | | | |
| Nurse printed name present | 500 | 497 | 3 | 0.6 |
| Nurse printed name correct | 500 | 500 | 0 | 0 |
| Nurse signature present | 500 | 500 | 0 | 0 |
| Nurse signature date present | 500 | 474 | 26 | 5.5 |
| **Total invalid** | **500** | **473** | **27** | **5.4** |
| **Consent to store samples for future research correct** | 500 | 487 | 13 | 2.7 |
| **IC had already been corrected in the past** | 500 | 64 | 436 | 12.8 |

* For those variables two fields were required in the IC form.

In 4.2% (21/ 500) of cases, the checkbox confirming that the participant/guardian/witness had been well informed about the study and had received an information sheet was left unchecked.

Among the 500 selected ICs, 5.2% (26/ 500) participants were minor in age. Signature of the legal guardian was missing in 11.5% (3/ 26). Among the 4.0% (20/ 500) illiterate participants, in 12.5% (5/ 40) the thumbprint, in 5% (1/ 20) the name or signature of an impartial witness and in 25.0% (5/ 20) the signature date of the impartial witness was missing.

In the personnel-related elements, in 0.6% (3/ 500) the name of the nurse obtaining the IC and in 5.2% (26/ 500) the date of the nurse's signature was not present. A total 5.4% (27/ 500) of ICs were invalid and participants' data were excluded from the final analysis.

In the IC section on agreeing to have samples stored for future research, in total 2.7% (13/ 500) were erroneous due to missing (2) or incorrect participant printed name (4), missing signature (6) or missing check in the box (5). Samples from participants who did not consent were consequently destroyed.

### Characteristics of study staff

To gain a better understanding of the role the study staff played in the adherence to GCP, the quality of data collection and the overall research outcomes, we conducted interviews with all staff involved (N = 30) (S4 Table). The median age was 31 years old (IQR: 29–34) and most were female (76.7%). A third (33.3%) reported to be the main contributor of the household income, with a few staff members executing an additional job (13.3%). In terms of educational background and study-relevant language skills, all study staff had a university degree and regular reading in French was common amongst them (86.2%). Half of them felt comfortable writing in French at a good (33.3%) or professional level (20.0%), while 46.7% reported less confident written language skills in French. In terms of GCP training, all except one had attended multiple trainings in their lifetime, and 50% had attended at least four trainings. Due to changes in the study staff, not all employees had taken part in the four training courses during the study period.

### Framework analysis of ISIs and FGDs

**Understanding of GCP.** Overall, GCP was perceived as a positive concept by the participants of the FGDs and ISIs. Furthermore, it was understood as an entry requirement for a clinical trial position and necessity to access a convenient and well-paid local employment. Staff members understanding of details of the concept varied and was closely related to their professional function in the study. Nurses and midwives understood that signatures and explanations of the IC, the implementation of quality control, as well as benefits for the patient are key elements of GCP. In contrast, more technical study personnel, such as data entry clerks, understood GCP as the concept of data reliability. Following standardized operating procedures (SOPs) and the importance of protecting confidentiality within the clinical trial were seen as important elements of GCP across all job functions.

**Perception of clinical trial documents and their usage.** The employees recognized that the use of quality control measures and trial-specific SOPs reduce the occurrences of errors.

As many of the study staff perceived the compliance with trial documents and procedures as a key element of GCP, they also thematized the IC process as well as the quality control. Employees working directly with patients described concerns about the procedures and credibility of the study among women interested in participating in the trial before going through the IC process with the study personal: *"[…] they're already worried that there might be rumors, for example, about child trafficking"* as reported by ID 05, Midwife. Such rumors increased the need of fact-based communication about the context and procedures of the study, from study personal to the participants. Midwives had also experienced that some women withdrew from the study before any blood samples were taken, as they assumingly were afraid of the procedure. The study staff acknowledged that handling of vulnerable groups such as minors and illiterates needed extra

precaution. Nevertheless, while the required GCP procedures of obtaining IC from vulnerable participants were reportedly followed, it was also perceived as an additional time investment by the study personal.

*What was important in the consent process […] we asked her to sign, as an adult, who knew how to read. If she was illiterate, we needed a witness who could read and write. If she was a minor, she needed a guardian to sign. If she was a minor and illiterate, she needed a witness and a guardian […] we had to explain even if we wasted time, sometimes we wasted a lot of time explaining things to someone - ID 04, Paramedic*

Along with the time spent on explaining the procedures, a high overall workload within the trial in general was commonly mentioned. Especially the documentation of the study procedures was seen as time-consuming: "[…] *there was a lot of paperwork to fill in"* (ID 30). Interestingly, two out of five employees that described quality control as an important element of GCP, also found it time-consuming. One suggested to minimize quality control measures within the study:

*And there again, perhaps I'm in favor of the person asking me that quality control is a time delay and I'm in favor of it too, perhaps we can minimize it in our study- ID 15, Data entry*

Non-compliance with quality control was reported to lead to problems within the study, as illustrated following:

*PIDs and TIDs don't match up sometimes. And there were times when I was picking it up because…about the samples for example, and maybe it was difficult especially for the children's samples […], and it was in the laboratory that we found (…) that we couldn't use (the sample) because it didn't follow the normal pattern, and (so) it was lost (to the study)- ID 13, Lab-Technician*

**Facilitators of GCP within the study.** Several staff stated that as applying GCP-compliant procedures became more routine, the previously described errors in quality control measures were reduced. In contrast to this common feedback, one midwife felt that over time she forgot, for example, how to perform quality control of the urine collection, mainly because of tiredness and a long commute to the study site (ID 02, Midwife). One of the main facilitators of GCP implementation mentioned by the respondents, was the repeated GCP training that the staff had received. The training was perceived as good and prepared them well in advance for contact with actual participants. Further, the repeated trainings reassured the staff in their activities. As one of the study nurses described it:

*When I started working here, there was good training, and it was often and I think it gave coherence to my work- ID 01, Nurse*

For one set of study staff, enhancing their personal competencies possibly leading to a career in research with potential studies abroad, was mentioned as a driver of the decision to work in a clinical trial. In contrast to that, for other employees, the main driver to work in the trial and participate in the GCP trainings was the 'convenience' of the position, which enabled them to work in their own region of residency.

**Barriers to GCP within the study.** One major barrier that emerged across all interviews and the FGD was the time pressure induced by the implementation of the GCP procedures and the adherence to the SOPs. The employees tried to follow procedures, but this added difficulties to the time management, as illustrated by this midwife's quote:

*It was the time that seemed, um, we're trying to manage the time because there were a lot of people but respecting consent, we have to take people one by one- ID 10, Midwife*

Amendments to the trial documents during the ongoing study, such as adapted SOPs or ICs made it difficult for study staff to understand and implement the adjusted procedures and worsened the problem of time management. Nearing expiry date of diagnostic material added a further component of time pressure. During phases with high incidence of COVID-19 infection, recruitment in the study centers was put on hold because employees reportedly had a fear of getting infected when in contact with study participants. Additionally, at times a lack of equipment at the local health facilities, such as body weight scales, added time pressure and stress for midwives, due to the fact that these tools needed to be fetched from another building.

**Recommendations to facilitate GCP adherence.** Respondents mentioned the differences based on financial contribution to the household income of the study staff and how this related to the motivation to work in clinical research. The sustainability of research capacities, as well as the need for adapted implementation of GCP in the Malagasy context were contextualized:

> I don't know if there will be any research in the future, professors and doctors in Madagascar will get together to discuss what good practice is more compatible here? because it's not the same, for example, abroad, if we talk about health and they're happy, but here at home even taking a step is money, time and all that- ID 19, Data Entry clerk

On the one hand, respondents recommended to pay the trial participants travel expenses, as was done within the freeBILy trial. On the other hand, they emphasized that participants should not receive monetary incentives to take part in the study activity itself, in reference to what they had been taught during the GCP certification. Long-term employment contracts for the study personal were recommended to reliably guarantee the continued adequate level of qualification needed. The educational heterogeneity among the study team, particularly in terms of the French language skills, led to problems in communication in between the study staff in some cases.

Several staff members recommended improving communication with study participants and communities within the study to avoid mistrust and keep transparency at a high level. Limiting the turnover of study personal, especially foreign researchers, and guaranteing consistency in contact persons on the side of the trial sponsor for the entire duration of the study was seen as an important aspect.

## Triangulation

For the purpose of interpretation, the quantitative findings from the analysis of the CRF and IC data entry fields, the structured survey of study staff and the qualitative findings from the analysis the data generated by FGDs and ISIs were charted against components of the conceptual framework, that is a) building factors, which include moral and scientific factors, and b) factors promoting the quality of clinical trials (S5 Table). In addition, the emerging results of this anaylsis suggest the extension of the existing de Pretto-Lazarova et al. framework to include external influences which do require monitoring and assessment and emerged to have vital interplay with the remaining framework components (Fig 3). The connecting elements and interactions betweeen the components that emerged from this analysis have been additionally ammended onto the framework to visualize the interconnection between building factors, assessement factors and promoting factors. One example of external influences impacting other trial aspects is represented by unexpected threats, such as the lockdown due to the COVID-19 pandemic, which had a major impact on the trial conduct of the freeBILy trial due to the interruption of trial activites during a shutdown. Capacity strengthening and sustainability of established capacities and infrastructure for potential integration into the healthcare system or future clinical research are essential elements in conducting trials in low-ressource settings. To note is the ambivalence of the requirement for rigorous documentation as one of the most vital element of clinical trial quality, but also as a factor which in freeBILy added significant workload on already streched study staff. Repeated training was reported as a key element of error reduction. Amendements to the protocol and study tools throughout the trial were pointed out as an additional source of errors and workload through updated documentation and required change of procedures and training.

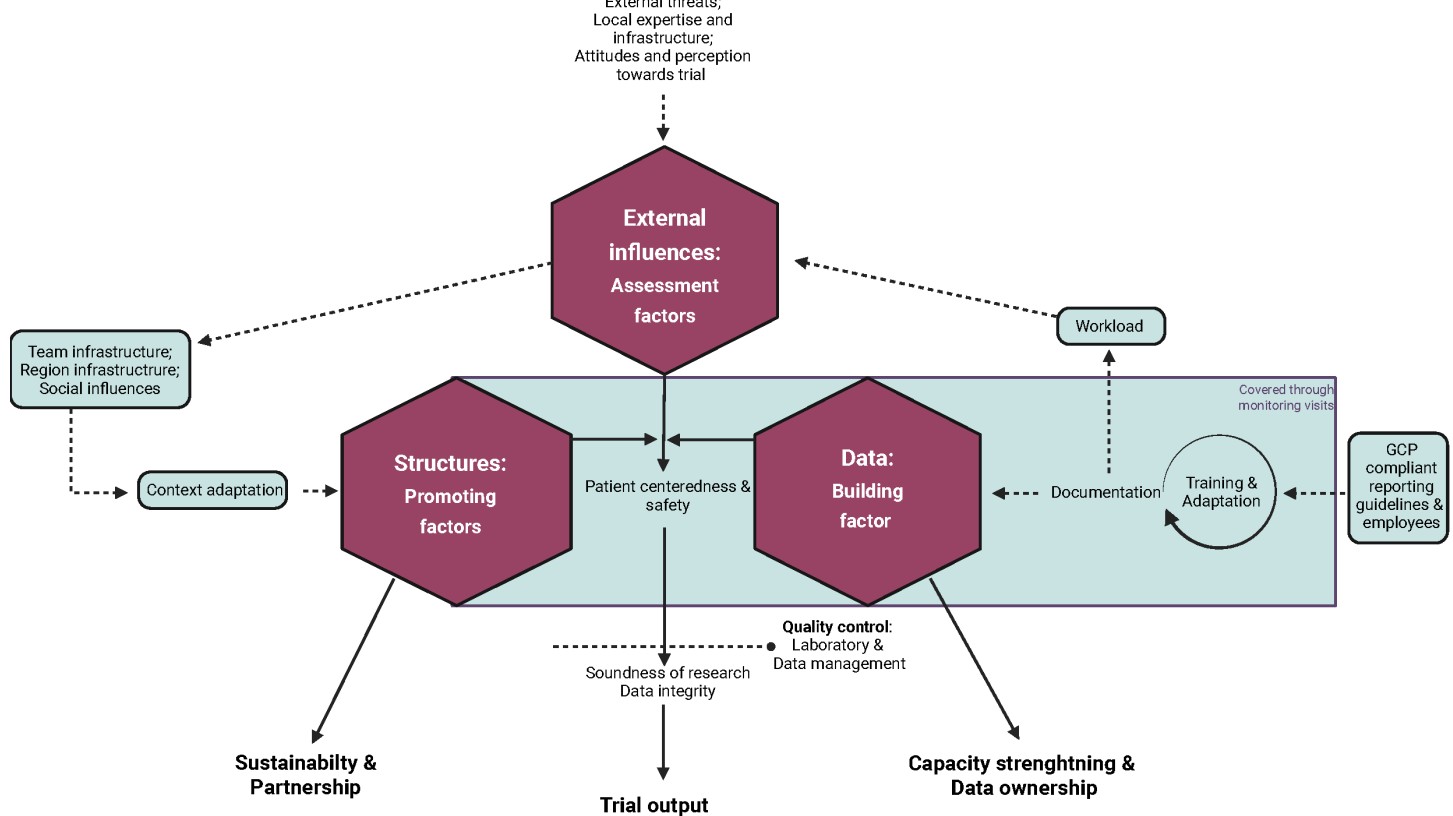

**Fig 3. Interrelated components of clinical trial quality in ressource-limited setting under implementation of GCP guideline, including factors outside of the normal trial monitoring: Major factors influencing the key elements of patient centredness and safety are displayed clockwise: a) Structures as promoting factors with the output of sustainability of the created infrastructure and establishment of longterm partnerships through team infrastructure, b) External influences as factors that require reoccuring assessment to control direct impact on trail mechnisms and safety, c) Data as the building factors of a clinical trial, together with the GCP compliant data collection bound to documentation and training cycles.** Together this leads to the output of strengthened professional and scientific capacity, while facilitating data ownership of local partners. In the sqaure box are the elements of a clinical trial covered by the required monitoring visit. Elements of quality control that ensure the integrity of the data and the soundness of the research in the laboratory and data management go beyond this monitoring. This means that even if all elements of monitoring have been fulfilled, the control of potential issues with sample collection and data management is considered essential for the reliable output of the clinical trial. Created in BioRender. Fusco, D. (2025) https://BioRender.com/8ia401e.

## Discussion

This study aimed to analyze barriers and facilitators for the implementation of GCP in resource-limited settings. Our results aim to inform the establishment of comprehensive research quality management in such contexts, through lessons learnt. We found that datacentric monitoring of GCP-conform studies neglects essential implementation elements, affecting not only data quality but also sustainability of local infrastructure and capacities. Ultimately this hampers the success of large scale and field-based research or clinical trials in low-resource settings.

This study combined findings of a datacentric analysis with a sociological perspective based on an analysis of FGDs and ISIs as well as a survey among study staff. These three components were combined in a final triangulation. This multi-method approach was chosen to provide guidance for a better implementation of GCP with the goal to improve robustness of future research studies and clinical trials in sub-Saharan African (SSA) countries.

With an overall rate of 1.8% incorrect data entries, the presented study reports a good quality of data recording according to a datacentric perspective and is well below the 5% of incorrect data entries defined as tolerable within studies conducted according to GCP [19]. Nevertheless, through the perspectives of the staff, it could be observed that external factors such as the working conditions, social components and infrastructure constraints impacted data quality. It clearly emerged that the workload of documentation was demanding and required a high level of quality control, especially during data entry. This contributed to negative perceptions among data management staff regarding data entry procedures and created a paradox whereby quality control was recognized as essential, yet simultaneously questioned due to workload constraints.

Despite the added workload, staff recommended more community-engagement like radio spots or community activities, as implemented in freeBILy. These patient engagement activities have shown to build trust and connection with the trial and ultimately facilitate compliance with the trial procedures through putting the community at the center [26]. However, as nurses worked not exclusively for the clinical trial, side studies and engagement activities further contributed to the difficult establishment of routines.

Additionally, the frequent change in study personal on the sponsor's side across the study period of four years represented a barrier in the overall study team's communication and trust. Under the light of equitable partnerships these dynamics in transnational partnerships need to be carefully monitored and should be adjusted [27]. Adequate and durable research funding and staff contracts should be ensured to cover an entire study from end to beginning.

Our findings underscore that achieving optimal data quality requires a delicate balance between staff understanding of GCP principles, their commitment to the study objectives, trust and manageable workload within the research team. To ensure continuity and sustainability, researchers must address staff fatigue while maintaining procedural compliance, as frustration from increased workloads can compromise adherence to protocols. Context-specific and team-tailored individual- and institutional capacity strengthening approaches have been recommended to be implemented from the early planning phase on [27,28]. By including mentoring programs alongside traditional training methodologies and assessments of critical infrastructure, sustained GCP implementation can be achieved.

Following the perception of difficulties to obtain consent for vulnerable groups (i.e., minors or illiterates), the deviations in respective ICs were particularly high: 11.5% signatures of guardians missing, 5% of signatures for impartial witness missing, and 12.5% of thumbprints missing. One aspect that might contribute to the missing signatures of the guardians is the assumed high emancipation of pregnant minors, especially in the context of a pregnancy service [29]. This aspect of IC given by emacipated minors has recently found more detailed specification in the new E6 (R3) ICH-GCP guideline, although it remains vague [7]. Country specific regulations need to be always monitored when obtaining IC from emancipated minors. Moreover, a properly completed IC may be technically correct; still, it does not guarantee that the participant comprehends its content, which is a key component of ethical research [30]. A review among SSA countries underscored the lack of comprehension among participants regarding study risks or side-effects of medication [31]. This aspect in particular cannot be investigated in a data centric approach and the need for additional clarification within the IC process to counteract misunderstandings and rumours became evident in the analysis of FGDs and ISIs.

We found that regular refresher training and increased familiarity with study procedures improved data quality over time. This was evident by the decreasing rate of weekly data entry errors over the study period, despite expanding CRF variables and complexity in later study timepoints. However, often underlying issues with data quality cannot be identified when only applying a datacentric viewpoint [32]. Staff interviews revealed that the dynamic nature of large, long-term trials undermined training benefits by requiring continuous adaptation to evolving protocols. Theory based study designs following implementation science principles might facilitate a clear introduction of changes through a systematic approach, mitigating these effects [33].

The errors in CRF data entry forms leading to subsequent correction, as well as the 12.8% of ICs corrected during the course of the study, had been identified as a result of regular reviews. This underscores the necessity for periodic field visits and study document checks beyond the required monitoring visits.

### Strengths and limitations

The novel combination of databank analysis, study staff survey and framework analysis of ISIs and FGDs, the large sample size of the initial clinical trial and the variety of viewpoints provided, from primary health care employee to laboratory worker and data clerk, represent the strengths of the present study. Despite these strengths this study is not without limitations.

Firstly, the strategy chosen to quantify the extent of deviations throughout the study included only those entries that differed in the REDCap records. Those that went undetected in the field were not included, suggesting that the true rate of incorrect data entries may be higher. Secondly, this research only included a randomly selected representative subset of the full dataset, and not all CRFs and IC forms were reviewed. Thirdly, due to the limited number of study staff, we were not able to draw any quantitative conclusion about the impact of different language levels or other discriminant factors on the quality of data collection which impacts generalizability. Self-reported skill level is subject to information bias, and social desirability bias in focus group settings cannot be entirely diminished. Furthermore, it would have been advantageous to test staff knowledge on GCP prior and after training sessions to evaluate their progression and the effectiveness of training. This study did not fully explore the individual definition of GCP of the study personnel besides the exploration of personal understanding of GCP presented in the qualitative analysis section. Nevertheless, due to the commonly conducted ICH GCP training the definition after the ICH GCP was applied within this analysis. Last but not least, through emergence in the qualitative section of this study, it could be seen that comprehension of the IC by the participant was a concern voiced by study staff. Due to the lack of data from the participant perspective this aspect could not be followed up more in depth in the scope of this study.

### Conclusion

This study shows the limitations of data-centric clinical trial management to assess GCP performances in the context of clinical trials in resource limited settings. Our findings highlight the importance of well-trained and integrated study staff as well as thorough preparation and context appropriate monitoring for high quality, patient centered and GCP-compliant research. By extending existing data-centric frameworks to include process-oriented aspects in the monitoring of clinical trials, outputs and reliability of study results can be maximized. More work is needed to build adapted standardized matrices that can allow a swifter and more contextualized assessment of study conducts. Capacity strengthening for local study staff should be a core component of any research project, as it fosters sustainable competency, strengthens livelihoods, and promotes long-term scientific advancement and partnerships in resource- limited settings.

### Supporting information

**S1 Table. Overview normalized data entry errors by timepoint.**
(DOCX)

**S2 Table. Incorrect data entry fields by type of variable.**
(DOCX)

**S3 Table. Incorrect data entry fields and outcome variables.**
(DOCX)

**S4 Table. Study staff characteristics.**
(DOCX)

**S5 Table. Framework matrix.**
(DOCX)

**S6 Table. Charting of quantitative and qualitative finding for triangulation.**
(DOCX)

## Acknowledgments

We firstly want to acknowledge and thank Prof. Raffaella Ravinetto for reading and revising the manuscript and providing insightful inputs.

The freeBILy consortium consists of the following: Leiden University Medical Center (LUMC), the Netherlands: Dr. G.J. van Dam;Dr. P.L.A.M. Corstjens; C.J. de Dood; P.T. Hoekstra, MSc; Dr. A.S. Amoah; and Dr. M.I. Keshinro.

Eberhard-Karls-Universität Tübingen (EKUT), Germany: Dr. A. Kreidenweiss

Bernhard-Nocht-Institut für Tropenmedizin (BNITM), Germany: Dr. N. G.Schwarz; Dr. D. Fusco; Dr. P. Klein; A. Jaeger; and Dr. E. Lorenz

Centre de Recherches Médicales de Lambaréné (CERMEL), Gabon: Dr. A.A.Adegnika; Dr. Y.J. Honkpehedji; Dr. J.C. Dejon-Agobe; R. Beh Mba; M. MbongNgwese; M. Nzamba Maloum; A. Nguema Moure; and B. Meulah T

Université de Fianarantsoa (UF), Madagascar: Dr. R. A. Rakotoarivelo; Dr. A.Ralaizandry; and Dr. M. Radomanana

Université d'Antananarivo (UA), Madagascar: Dr. R. Rakotozandrindrainy; Dr. N.Rakotozandrindrainy; Dr. Marie Jean-nine Solonirina; and Dr. J. Randriamanjara

Centre d'Infectiologie Charles Mérieux (CICM), Madagascar:Dr. M. Rakoto Andrianarivelo; Dr. T. Rasamoelina; Dr. R. Razafindrakoto;

Privada Instituto de Salud Global Barcelona (ISGLobal), Spain: Dr.E. Sicuri and C. Aerts, PhD

## Author contributions

**Conceptualization:** Leonard Gunga, Pia Rausche, Natalie Fischer, Daniela Fusco.

**Data curation:** Leonard Gunga, Pia Rausche, Nagham Issa, Anna Jaeger.

**Formal analysis:** Leonard Gunga, Pia Rausche, Nagham Issa, Natalie Fischer.

**Funding acquisition:** Rivo Andry Rakotoarivelo, Mandranto Rasamoelina, Govert van Dam, Norbert Schwarz, Jürgen May, Raphaël Rakotozandrindrainy, Daniela Fusco.

**Investigation:** Leonard Gunga, Rivo Andry Rakotoarivelo, Mandranto Rasamoelina, Jeannine Solonirina, Elveric Fesia, Ravo Razafindrakoto, Njary Rakotozandrindrainy, Mickael Radomanana, Valentina Marchese, Jean-Marc Kutz, Aaron Remkes, Raphaël Rakotozandrindrainy, Natalie Fischer, Daniela Fusco.

**Methodology:** Daniela Fusco.

**Project administration:** Leonard Gunga, Raphaël Rakotozandrindrainy, Natalie Fischer, Daniela Fusco.

**Supervision:** Dewi Ismajani Puradiredja.

**Validation:** Natalie Fischer, Daniela Fusco.

**Visualization:** Pia Rausche.

**Writing – original draft:** Leonard Gunga, Pia Rausche, Natalie Fischer, Daniela Fusco.

**Writing – review & editing:** Rivo Andry Rakotoarivelo, Mandranto Rasamoelina, Jeannine Solonirina, Elveric Fesia, Ravo Razafindrakoto, Njary Rakotozandrindrainy, Mickael Radomanana, Valentina Marchese, Nagham Issa, Jean-Marc Kutz, Aaron Remkes, Anna Jaeger, Dewi Ismajani Puradiredja, Govert van Dam, Norbert Schwarz, Jürgen May, Raphaël Rakotozandrindrainy.

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
