## [Decision Letter · Decision Letter 0]

28 Oct 2025

PNTD-D-25-01527Implementation of good clinical practice in clinical research in the context of limited resources settings: lessons learnt from the freeBILy trial using a multi-methods approach PLOS Neglected Tropical Diseases Dear Dr. Fusco, Thank you for submitting your manuscript to PLOS Neglected Tropical Diseases. After careful consideration, we feel that it has merit but does not fully meet PLOS Neglected Tropical Diseases's publication criteria as it currently stands. Therefore, we invite you to submit a revised version of the manuscript that addresses the points raised during the review process. Please submit your revised manuscript within 30 days Dec 27 2025 11:59PM. If you will need more time than this to complete your revisions, please reply to this message or contact the journal office at plosntds@plos.org. Please include the following items when submitting your revised manuscript: * A rebuttal letter that responds to each point raised by the editor and reviewer(s). You should upload this letter as a separate file labeled 'Response to Reviewers'. This file does not need to include responses to any formatting updates and technical items listed in the 'Journal Requirements' section below. * A marked-up copy of your manuscript that highlights changes made to the original version. You should upload this as a separate file labeled 'Revised Manuscript with Track Changes'. * An unmarked version of your revised paper without tracked changes. You should upload this as a separate file labeled 'Manuscript'. If you would like to make changes to your financial disclosure, competing interests statement, or data availability statement, please make these updates within the submission form at the time of resubmission. Guidelines for resubmitting your figure files are available below the reviewer comments at the end of this letter. We look forward to receiving your revised manuscript. Kind regards,Anna Giné-MarchAcademic EditorPLOS Neglected Tropical Diseases

Gabriel Rinaldi

Section EditorPLOS Neglected Tropical Diseases

Shaden Kamhawi

co-Editor-in-Chief

Paul Brindley

co-Editor-in-Chief

**Additional Editor Comments:**The study addresses a topic of high public health relevance, providing valuable insights into the implementation of Good Clinical Practice (GCP) in resource-limited settings. Some revisions are recommended to strengthen the manuscript. It would be helpful to further clarify how the results support the conclusions and to emphasize the implications for policy and future clinical trials in resource-limited settings.**Journal Requirements:**

1) Please provide an Author Summary. This should appear in your manuscript between the Abstract (if applicable) and the Introduction, and should be 150-200 words long. The aim should be to make your findings accessible to a wide audience that includes both scientists and non-scientists. Sample summaries can be found on our website under Submission Guidelines:

4) In the online submission form, you indicated that The datasets used and analyzed during the current study are available from the corresponding author on reasonable request and after validation of the freeBILy steering committee.. All PLOS journals now require all data underlying the findings described in their manuscript to be freely available to other researchers, either

1. In a public repository

2. Within the manuscript itself

3. Uploaded as supplementary information.

5)  Please ensure that the funders and grant numbers match between the Financial Disclosure field and the Funding Information tab in your submission form. Note that the funders must be provided in the same order in both places as well.

**Reviewers' comments:**Reviewer's Responses to Questions

**Key Review Criteria Required for Acceptance?**

**Methods**

-Are the objectives of the study clearly articulated with a clear testable hypothesis stated?

-Is the study design appropriate to address the stated objectives?

-Is the population clearly described and appropriate for the hypothesis being tested?

-Is the sample size sufficient to ensure adequate power to address the hypothesis being tested?

-Were correct statistical analysis used to support conclusions?

-Are there concerns about ethical or regulatory requirements being met?

Reviewer #1: Dear,

I am honored to contribute to the review of this manuscript. Please find below a set of comments I consider pertinent:

Sampling and Bias

Please provide detailed information on the randomization of the subsets: sampling design, the software used (and the random seed), and a step-by-step description of the selection procedure. Justify the sample size and assess representativeness by comparing the sample to the full cohort.

Definitions and Error Metrics

Please clarify the operational definitions: what constitutes a final discrepancy between the CRF and the source? How are mandatory fields handled when missing? I recommend adding a severity classification (critical/major/minor) and stratifying errors by variable type and timepoint.

Informed Consent (IC)

Adopt objective criteria to classify invalid IC and specify the analytical impact (e.g., exclusion in the primary analysis and inclusion in a sensitivity analysis). Align the discussion with current guidelines (e.g., ICH-E6(R3).

Triangulation and Integration

Include a summary table linking quantitative and qualitative findings to operational recommendations (who is responsible, when actions occur, and feasibility/cost).

Language Review

I recommend an English language edit (American English), with consistent terminology and improvements to clarity and flow.

Reviewer #2: The overall study design must be stated rest minor modifications in methods section.

Reviewer #3: -Are the objectives of the study clearly articulated with a clear testable hypothesis stated?

= the objectives are clearly stated in lines 99-107

-Is the study design appropriate to address the stated objectives?

== Implementing a quantitative, datacentric approach along with a qualitative structured interviews is a nice approach. However, I am uncertain if data entry errors would be considered GCP non-compliance? Wouldn’t data corrections (corrected appropriately) indicate adherence to GCP?

== Lines 214-218: It would be nice to include more details on how you approached the triangulation

==Line 160: “the results from the manually reviewed informed consent forms” – but it is unclear who manually reviewed them?

-Is the population clearly described and appropriate for the hypothesis being tested?

==Yes, the population is clearly described and appropriate for the hypothesis (mothers from the freeBILy study). However a few items remained unclear to me:

== Line 166 and 167: The authors have not yet introduced the term “ISIs” and “FGD”, I would recommend providing the full name here in the first instance (unless the list of abbreviations is meant to solely address this?)

==Line 167: it’s unclear what authors mean by “15 purposively selected employees for information rich cases” ?

== Lines 167-169: Could the authors elaborate on why 15 staff were chosen for one set of interviews and 15 were chosen for the other? Were they randomly assigned? and what was the difference between the two interviews?

== Lines 185-187: were missing fields considered as data entry errors?

==Lines 185-187: If double data entry was conducted, were the records merged in REDCap and then the errors analyzed? Or were the errors considered per data entry staff, for example?

==Lines 194: Perhaps the authors would like to clarify “consents” (i.e. 500 consents were chosen)?

-Is the sample size sufficient to ensure adequate power to address the hypothesis being tested?

==I do not see a problem with the sample size, but just a few comments here:

== Line 182: 500 participants, out of 4612 participants, should be 10.8% of the total number of participants, not 11.2%

== Line 182: participants were “randomly selected” but could the authors elaborate on how this was done?

-Were correct statistical analysis used to support conclusions?

==The authors used the Mann-Kendall Test, which appears appropriate for the data being analyzed

-Are there concerns about ethical or regulatory requirements being met?

== No concerns about ethical or regulatory requirements. Details surrounding the ethical considerations are provided in lines 220-227.

**Results**

-Does the analysis presented match the analysis plan?

-Are the results clearly and completely presented?

-Are the figures (Tables, Images) of sufficient quality for clarity?

Reviewer #1: They are suitable.

Reviewer #2: Some calculations has to be redone as mentioned in the reviewer attachment.

Table 1 needs to be redone with correct calculations.

Reviewer #3: -Does the analysis presented match the analysis plan?

==The analysis presented matches the analysis plan; however, I have one comment:

==Line 290: how did the authors decide which IC errors were minor or major?

-Are the results clearly and completely presented?

==Line 411: Could the authors elaborate more on the triangulation results? This seems to me to be the keystone to the work (bringing together two aspects of the analysis), but I struggle to conceptualize the results.

==The authors may want to elaborate on how / if the study team changed throughout time, and if this had an impact on data quality

==Lines 281: It would be helpful to know if the IC errors were corrected & how?

-Are the figures (Tables, Images) of sufficient quality for clarity?

== The tables throughout are sufficient quality and clarity. However, in my opinion, Figure 3 is difficult to follow

**Conclusions**

-Are the conclusions supported by the data presented?

-Are the limitations of analysis clearly described?

-Do the authors discuss how these data can be helpful to advance our understanding of the topic under study?

-Is public health relevance addressed?

Reviewer #1: They are suitable.

Reviewer #2: -The conculations are mentioned appropriately.

-Limitations properly addressed.

-Public health relevance is mentioned correctly.

Reviewer #3: -Are the conclusions supported by the data presented?

==The authors’ conclusion is that interpreting datacentric results in the context of GCP is an insufficient approach in resource-limited settings. The authors indicate that training and context appropriate monitoring is paramount to ensure high quality and compliant research. However, I find it difficult to see succinctly how these conclusions were made. Perhaps if the authors could elaborate on the triangulation aspect of the results, it would become clearer. The data entry errors were very small overall, despite limited resources and extensive paperwork.

-Are the limitations of analysis clearly described?

==The limitations are explored in lines 526-541.

-Do the authors discuss how these data can be helpful to advance our understanding of the topic under study?

== Yes

-Is public health relevance addressed?

==I think this point could be strengthened.

**Editorial and Data Presentation Modifications?**

Reviewer #1: (No Response)

Reviewer #2: The submitted research article has public health relevance. May be accepted after minor revisions if properly addressed.

Reviewer #3: I have a few general comments for the authors:

==Line 72: It could be helpful to provide a reference for E6 R2

==Line 81: Elaborating on how E6(R3) has shifted clinical trial management into the right direction could guide readers towards what are still limiting factors

==Line 138: REDCap requests specific citations when their database is used (additional information located on their website under “Resources  citations”). Authors should consider including this (https://doi.org/10.1016/j.jbi.2008.08.010 and https://doi.org/10.1016/j.jbi.2019.103208).

==Line 234: A full-stop should be used instead of a comma when reporting percentages (i.e. 1.8%)

==I am not sure if it's because of the 'justify' centering of the text, but there are a few instances where it appears there are 2 spaces between words (e.g. Line 274, 279, 216, 395)

==Line 338: I believe the authors meant to write 'midwives'

**Summary and General Comments**

Reviewer #1: Dear,

I am honored to contribute to the review of this manuscript. Please find below a set of comments I consider pertinent:

Sampling and Bias

Please provide detailed information on the randomization of the subsets: sampling design, the software used (and the random seed), and a step-by-step description of the selection procedure. Justify the sample size and assess representativeness by comparing the sample to the full cohort.

Definitions and Error Metrics

Please clarify the operational definitions: what constitutes a final discrepancy between the CRF and the source? How are mandatory fields handled when missing? I recommend adding a severity classification (critical/major/minor) and stratifying errors by variable type and timepoint.

Informed Consent (IC)

Adopt objective criteria to classify invalid IC and specify the analytical impact (e.g., exclusion in the primary analysis and inclusion in a sensitivity analysis). Align the discussion with current guidelines (e.g., ICH-E6(R3).

Triangulation and Integration

Include a summary table linking quantitative and qualitative findings to operational recommendations (who is responsible, when actions occur, and feasibility/cost).

Language Review

I recommend an English language edit (American English), with consistent terminology and improvements to clarity and flow.

Reviewer #2: Overall study is important but the paper flow and redaction needs improvement. The calculations has some mistakes which needs to be addressed. If the concerns are properly worked on than paper may be accepted after minor revision. research ethics were followed and proper permissions were taken as per the article.

Reviewer #3: This manuscript is very well written, thorough, and clearly described. I appreciate the authors work on something so crucial: identifying challenges and highlighting the need for pragmatism when conducting clinical trials in resource-limited settings. My main comment is that using data entry errors as a 'semi-proxy' for GCP compliance does not completely address the entire aspect of GCP. I would rather say, were the errors properly corrected, as data entry errors will occur in all settings. It would be interesting to compare, perhaps, deviations related to inclusion/exclusion criteria, prohibited medications, etc., and how / if the frequency of these deviations changed after training sessions or monitoring visits. In the same vein, it would be helpful if the authors compared how these results might differ from clinical trials in other resource-sufficient settings. I would also be interested to read how the conclusions from this project could be used moving forward (policy or in other trials, for example).

PLOS authors have the option to publish the peer review history of their article (what does this mean?). If published, this will include your full peer review and any attached files.

Reviewer #1: No

Reviewer #2: **Yes:** Dr. Ambreen Fatema

Reviewer #3: No

**Figure resubmission:**While revising your submission, we strongly recommend that you use PLOS’s NAAS tool (https://ngplosjournals.pagemajik.ai/artanalysis) to test your figure files. NAAS can convert your figure files to the TIFF file type and meet basic requirements (such as print size, resolution), or provide you with a report on issues that do not meet our requirements and that NAAS cannot fix.

After uploading your figures to PLOS’s NAAS tool - https://ngplosjournals.pagemajik.ai/artanalysis, NAAS will process the files provided and display the results in the 'Uploaded Files' section of the page as the processing is complete. If the uploaded figures meet our requirements (or NAAS is able to fix the files to meet our requirements), the figure will be marked as 'fixed' above. If NAAS is unable to fix the files, a red 'failed' label will appear above. When NAAS has confirmed that the figure files meet our requirements, please download the file via the download option, and include these NAAS processed figure files when submitting your revised manuscript **Reproducibility:**

---

## [Editor Report · Decision Letter 1]

28 Dec 2025

Dear Dr. Fusco,

We are pleased to inform you that your manuscript 'Implementation of good clinical practice in clinical research in the context of limited resources settings: lessons learnt from the freeBILy trial using an embedded mixed methods approach' has been provisionally accepted for publication in PLOS Neglected Tropical Diseases.

Best regards,

Anna Giné-March

Academic Editor

Gabriel Rinaldi

Section Editor

Shaden Kamhawi

co-Editor-in-Chief

Paul Brindley

co-Editor-in-Chief

---

## [Editor Report · Acceptance letter]

Dear Dr. Fusco,

We are delighted to inform you that your manuscript, "Implementation of good clinical practice in clinical research in the context of limited resources settings: lessons learnt from the freeBILy trial using an embedded mixed methods approach," has been formally accepted for publication in PLOS Neglected Tropical Diseases.

Best regards,

Shaden Kamhawi

co-Editor-in-Chief

Paul Brindley

co-Editor-in-Chief
